# A Brief History of Single-Particle Tracking of the Epidermal Growth Factor Receptor

**DOI:** 10.3390/mps2010012

**Published:** 2019-01-30

**Authors:** David T. Clarke, Marisa L. Martin-Fernandez

**Affiliations:** STFC Central Laser Facility, Research Complex at Harwell, Rutherford Appleton Laboratory, Didcot OX11 0QX, UK; dave.clarke@stfc.ac.uk

**Keywords:** single molecule tracking, epidermal growth factor receptor, cell signaling, protein–membrane interactions, oligomerization, endocytosis

## Abstract

Single-particle tracking (SPT) has been used and developed over the last 25 years as a method to investigate molecular dynamics, structure, interactions, and function in the cellular context. SPT is able to show how fast and how far individual molecules move, identify different dynamic populations, measure the duration and strength of intermolecular interactions, and map out structures on the nanoscale in cells. In combination with other techniques such as macromolecular crystallography and molecular dynamics simulation, it allows us to build models of complex structures, and develop and test hypotheses of how these complexes perform their biological roles in health as well as in disease states. Here, we use the example of the epidermal growth factor receptor (EGFR), which has been studied extensively by SPT, demonstrating how the method has been used to increase our understanding of the receptor’s organization and function, including its interaction with the plasma membrane, its activation, clustering, and oligomerization, and the role of other receptors and endocytosis. The examples shown demonstrate how SPT might be employed in the investigation of other biomolecules and systems.

## 1. Introduction

Single molecule (SM) imaging methods have been transformational in our understanding of the functioning of complex biological systems [1]. By allowing us to see beneath the ensemble average, these techniques can probe individual molecules in highly heterogeneous systems. This means that rare events can be identified and studied, and detailed information can be extracted without the requirement to synchronize the behavior of large numbers of molecules. In addition, the ability to localize individual molecules with a precision much better than optical resolution allows us to probe the architecture of cellular structures and molecular assemblies, which is inaccessible via other techniques.

In the relatively short time that SM methods have been developed and applied to biological research, a vast range of systems have been studied using a wide range of methodologies. We do not intend this article to be a full review of single-particle tracking (SPT) techniques, but instead focus on a specific research area and use it as an exemplar of how the development of SM techniques of increasing sophistication has provided detailed insights into the structure and function of biological molecules. It is hoped that this will inspire life science researchers to consider how SPT methods might be useful for their particular research areas. For those wishing to understand SPT at a fundamental level, more comprehensive reviews describing the evolution and current state of the art are available, e.g., [2]. We have excluded from our definition of SPT so-called “super-resolution” microscopy methods that can be used to construct images from the localization of many single molecules, as we consider that these are best discussed in the context of other microscopy techniques, which are outside the scope of this article. Again, a number of reviews of these methods can be found in the literature, e.g., [3].

Here, we concentrate on the group of methods that can be referred to as single-particle tracking SPT, in which the fluorescence of individual molecules of interest is tracked spatially and/or temporally in biological samples, in particular mammalian cells in culture [2,4,5]. In short, single-molecule tracking in cells involves labeling molecules of interest with a fluorescent marker. This can be typically achieved either by conjugating an organic fluorescent dye or quantum dot to a biomolecule that binds to the target molecule specifically (either the target molecule’s natural ligand or an antibody or antibody-like molecule), or by genetically modifying the cells to express a fluorescent protein fused with the target molecule [6]. The cells are then imaged in a fluorescence microscope and recorded on a camera. The intensity and position of diffraction-limited fluorescence spots detected from labeled single molecules are recorded in a time-resolved manner for subsequent analysis.

The system we have selected as a model is the epidermal growth factor receptor (EGFR), which is a transmembrane glycoprotein that belongs to the super-family of receptor tyrosine kinases (RTKs). The EGFR family consists of four homologous members (EGFR, HER2, HER3, and HER4) and regulates the signal transduction processes that are involved in cellular proliferation, survival, differentiation, function, and motility [7,8]. EGFR is frequently hyperactivated in human cancers via mutation and/or overexpression [9]. This driving role in malignancy has made EGFR a key target for anti-cancer therapy [10].

The EGFR has been well-characterized structurally, and a model for receptor activation in which inactive monomeric EGFR dimerize on ligand binding has been established for some time [11,12]. However, it has become increasingly evident that structure alone cannot answer a number of questions concerning the functioning of the receptor in vivo. In particular, areas in which information from other methods has been required include the role of higher order oligomers, clustering of the receptors in cells, interactions of the receptor with the plasma membrane, and interactions between different members of the receptor family and with other receptors. Unsurprisingly, researchers have turned to SPT methods to attempt to shed light on some of these areas. The EGFR family is highly amenable to study using SM methods, as it is located in the plasma membrane, and can therefore be studied using total internal reflection fluorescence (TIRF) microscopy. A number of fluorescence labeling strategies are available for the EGFR, including the labeling of one if its ligands, which is most commonly epidermal growth factor (EGF), antibodies and antibody-like molecules, and fusion with fluorescent proteins. Given this suitability, EGFR was one of the first systems to be studied using SM tracking, and here, we show how the method has been used to study the receptor over more than two decades, and how the knowledge obtained has increased as SPT methods have developed (for examples of SM tracking of other members of the EGFR family, the reader is referred to [13,14,15,16]). The EGFR story provides a useful illustration of how SPT might be applied to other biological molecules of interest.

## 2. Background to Single-Particle Tracking Techniques

In order to understand how SPT methods have contributed to the understanding of the structure and dynamics of the EGFR, it is first necessary to introduce a few concepts that underpin the techniques. Firstly, the molecules of interest must be labeled with a fluorescent probe so that they can be visualized in the single molecule microscope. The fluorescent label can be a fluorescent protein (FP), a fluorescent dye, or a quantum dot (QD). Considerations of which type of label to use are influenced by a number of factors, in particular the delivery of the label to the target protein, and the photophysical properties of the probe. Probe targeting can be through the labeling of a molecule with a high affinity for the target, for example a ligand, antibody, or antibody-like molecule. In general, antibody fragments or small antibody-like molecules such as nanobodies or affibodies are preferred, as the relatively large antibody molecules may impede or alter the motion of the target molecule [2]. Photophysical effects include photobleaching, in which the fluorescence of the probe is lost due to bond breakage or chemical reaction, while the probe is in an excited state following the absorption of a photon, or blinking, in which fluorescence is temporarily lost as the probe switches from a radiative to a non-radiative relaxation pathway [17]. It is beyond the scope of this article to provide a comprehensive guide to probe selection, but a number of factors have to be considered. These include probe brightness (QDs are brighter than organic dyes), the ease of labeling (FPs do not require the use of an external probe such as an antibody or ligand), experiment duration (QDs maintain fluorescence emission for much longer than organic dyes), or perturbation of the system (QDs are large, and may interfere with some interactions). The tendency of some organic probes to attach to the substrate that is used for cell culture and imaging should also be considered [18]. Blinking characteristics are important for SPT, and the effect of blinking is discussed below. For a more detailed discussion of probe selection, see for example [19].

The second aspect to be considered is the instrument that is used to collect the SPT data. One of the main challenges when dealing with mammalian cells is the high level of background arising from autofluorescence. In SPT experiments on mammalian cells, the approach that is usually taken to minimize the autofluorescence background is to use TIRF. TIRF depends on the difference in refractive index between the coverslip on which the cells are cultured, and the medium in which the cells are maintained. The light that strikes the interface between them is totally internally reflected, resulting in an evanescent field that illuminates the basolateral surface of cells to a depth of around 100 nm from the coverslip, avoiding the excitation of background fluorescence from the cytoplasm [20]. Frequently, multiple illumination and emission wavelengths are employed, and the fluorescence emitted in the distinct spectral ranges is optically filtered and collected on different areas of the detector (e.g., [21]). Labeling different classes of molecules with different fluorescent probes allows the monitoring and quantification of interactions between them. An alternative approach to TIRF is to use so-called inclined illumination [22], which is also known as highly inclined laminated optical light sheet (HILO) microscopy. This method increases the illumination depth to around 500 nm, and can be used to illuminate the apical surface of cells. Usually, the detection of the fluorescence signals is by either an electron-multiplying CCD (EMCCD) or more recently, a Scientific CMOS (sCMOS) camera. A detailed treatment of TIRF microscopy theory and practice can be found in [23].

One of the most valuable features of SPT is the ability to locate the position of the particles with a precision better than the optical resolution of the microscope. The principle is that a single-particle emitter is imaged as the point spread function (PSF) of the microscope. By fitting the image with the PSF, it is possible to locate the position of the emitter. In practice, a Gaussian profile is frequently used instead of the true PSF, as this is easier computationally, and provides acceptable results. The localization precision is dependent on the number of photons detected and the size of the PSF, according to the equation:$$\Delta_{loc}=\frac{\Delta }{\surd N}$$
where Δ*_loc_* is the localization precision, Δ is the full-width half-maximum (FWHM) of the PSF, and ***N*** is the number of photons detected [24]. Typically in SPT experiments, the localization precision can range from a few nanometers to a few dozen nanometers. The first challenge in SPT data analysis is to identify and locate PSF-sized features against a residual background of fluorescence that cannot be entirely eliminated with the use of TIRF illumination. With bright fluorescence emitters, simple thresholding can be used, but for lower signal-to-noise ratios SNRs, more complex statistical methods such as Bayesian segmentation [25,26] or likelihood-based approaches [27] are often employed. These methods use a model of what a single particle feature is expected to look like, and determine the likelihood that a potential feature is consistent with that model. It is worth noting that the principles of feature detection and localization for SPT are identical to those for the detection of single molecules in localization-based super-resolution microscopy techniques such as photo-activated localization microscopy (PALM) and stochastic optical reconstruction microscopy (STORM). Therefore, methods developed for these imaging techniques can be applied more generally to the analysis of SPT data. One example is the application of methods that were originally developed for astronomy for single-particle detection in crowded fields of view [28].

Having detected and localized single particles, the next challenge for SPT is to track how their position and intensity changes during the course of an experiment. This enables the experimenter to determine the types of motion of molecules of interest, and multi-color SPT can be used to investigate the type, location, and duration of interactions between molecules. A number of examples of this are given below, where we describe the evolution of the use of SPT for studying EGFR. Obtaining single-particle tracks is not a simple matter of locating the particles at each time point and linking the positions together. Blinking means that particles may disappear for one or more frames in a data series. The tracks of molecules may come together or cross, then diverge, making the challenge one of identifying which trajectory forms part of a continuous track. Tracking methods generally attempt to overcome these difficulties by adopting a heuristics-based approach. One of the problems is that these methods tend to optimize for longer track lengths [29], being unable to satisfactorily distinguish one long track from a set of unconnected shorter ones. Statistical approaches have been taken to attempt to solve this problem [27,30]. In tracking as well as detection, there has also been crossover between SPT and localization-based super-resolution microscopy methods. The sptPALM technique uses photoswitchable fluorescent probes to activate multiple ensembles of molecules. This means that single-molecule tracks can be obtained at higher densities than possible with conventional tracking methods (up to ~50 per μm**^2^**) [31]. A detailed comparison of the performance of a number of tracking methods can be found in [32].

One of the most useful parameters that can be determined from single particle tracks is the mean squared displacement (MSD) of the particles. The MSD is an expression of the extent of space that a single particle explores as a function of the time since tracking begins. The MSD is defined by the generic formula:$$msd\left(\tau \right)=\langle \Delta r{\left(\tau \right)}^{2}\rangle =\langle {\left[r\left(t+\tau \right)-r\left(t\right)\right]}^{2}\rangle$$
where *r*(*t*) is the position of the particle at time *t*, and *τ* is the lag time between the two positions taken by the particle that is used to calculate the displacement *Δr*(*τ*) = *r*(*t* + *τ*) − *r*(*t*). By measuring MSD, it is possible to determine the nature of the particle’s motion, i.e., whether it is freely diffusing, restricted, or directed [2,4,33]. Below, we review how the measurement of MSD has allowed researchers to determine the nature of motion of the EGFR under a range of conditions. Once single-particle tracks have been obtained, they can also be used to extract information on the kinetics of the association and disassociation of molecules in cells (see for example, [34]). A common use of MSD is to obtain the diffusion coefficient of the molecule in question. This is a non-trivial problem because of the presence of different types of motion, and the relatively short lengths of the tracks that are typically obtained from fluorescent probes. For a detailed discussion of the calculation of the diffusion coefficient from SPT experiments, see for example [35]. Below, we show specific examples of how SPT has been used to extract kinetic parameters of interactions within the EGFR family.

Another single-molecule technique that has been applied to the EGFR is Förster resonance energy transfer (FRET), which has been used for many years as a “spectroscopic ruler” to measure separations between two fluorescent probes in the range of approximately two to eight nm [36]. This technique can also be applied in single-molecule mode to measure the distance between two individual fluorescent probes in real time. More details on single-molecule FRET (smFRET) can be found in the literature, e.g., [37]. The intensities of donor and acceptor fluorescence can be tracked simultaneously in spots where both are present. When FRET is occurring, anticorrelated intensity traces occur; increases in acceptor fluorescence occur simultaneously with decreases in donor fluorescence.

The final single-molecule technique that we discuss in the context of EGFR is fluorophore localization imaging with photobleaching (FLImP) [38]. FLImP relies on a combination of single-molecule localization and single-step photobleaching to map out with high precision the separation between molecules in a complex, in a range from ~5–100 nm. In short, FLImP depends on measuring the shift in the position of a PSF containing two or more fluorescence emitters when one of those emitters bleaches. By measuring the shift in position of the PSF on photobleaching, it is possible to determine the separation between bleached and emitting molecules. FLImP has been developed and refined over a number of years, and has been established as a useful method for the evaluation of EGFR oligomerization in cells [39]. Its effectiveness has depended on the development of statistical methods that allow the intermolecular separations to be quantified with accurate error analysis.

In the remainder of the article, we describe how the methods introduced above have been developed and applied to the study of a number of facets of EGFR structure and function.

## 3. Early Single-Particle Tracking Studies of Epidermal Growth Factor Receptor

The earliest SPT-based study of EGFR in cells was published by Kusumi et al. in 1993 [40]. Prior to this work, dynamics of receptors in membranes were studied using fluorescence photobleaching recovery (FPR) [41,42], which only reports ensemble behavior. This paper is essentially a proof-of-principle study demonstrating the possibility of tracking individual receptors in living cells. This work was done before the development of TIRF-based SPT, and used a technique known as nanovid microscopy [43] to track receptor motion. In this method, the target molecules are labeled with gold nanoparticles rather than fluorescent probes. In this case, EGFR’s ligand EGF was conjugated to 40-nm gold particles. Other molecules investigated in the same study were E-cadherin and transferrin receptor. This study reported four characteristic types of motion for the receptors, and demonstrated that SPT can be used to characterize the nature of the molecule’s movement within the plasma membrane. This is illustrated in Figure 1, which shows the types of motion tracks that can be observed. Figure 1a shows what can be described as “confined motion”, in which the molecules are confined to a small area of the membrane, and have a very restricted range of motion. Figure 1b shows what is known as “restricted motion”, in which the molecule is able to diffuse freely within a certain area, but does not leave its bounds during the course of the experiment. Figure 1c shows “simple diffusion mode”, in which molecules diffuse freely in the membrane, showing simple Brownian motion, and Figure 1d shows mainly directed motion, in which movement is largely unidirectional, with a low level of random motion superimposed. The presence of two or more types of motion within one track is common in biological systems. Information on motion type can be obtained by determining mean squared displacement (MSD) curves. The shape of the MSD curve depends on the type of motion, and Figure 1e–h shows MSD curves that would be obtained from the theoretical tracks shown in Figure 1a–d. Detailed quantifiable information may require sophisticated analysis algorithms to classify the different motion types correctly. Statistical approaches such as Bayesian segmentation have been used for this [44].

Kusumi et al. reported a majority of EGFR undergoing restricted motion, and concluded that this was due to confinement of the receptors by what they termed the “membrane skeleton fence”. This early study demonstrated the potential of SPT not only to allow the development of a model for the receptor’s interaction with the membrane, but also to provide quantitative information on diffusion rates.

Despite the valuable information yielded from the study described above, the use of SPT in cells remained low for a number of years, possibly because of the limitations of the nanovid microscopy technique and its requirement for labeling with relatively large gold nanoparticles. The advent of fluorescent-based SPT allowed much more sophisticated investigations of receptor behavior, starting with a seminal paper published in 2000 [46], in which Sako et al. demonstrated the power of using various SPT methods to measure a number of aspects of EGFR behavior in cells. In this study, which was carried out before the crystal structures of the EGFR became available [47,48], EGF was labeled with the fluorescent probe Cy3 and the fluorescent conjugate added to cultured A431 cells, which overexpress EGFR. The cells were imaged in a fluorescence microscope that was capable of TIRF and inclined illumination, and the intensity and position of fluorescent EGF were followed in both the basolateral and apical surfaces. The intensity of fluorescent spots was plotted versus time, showing step-like traces (Figure 2a). These steps are characteristic of single-molecule traces, and are caused by the photobleaching of individual molecules (“single-step photobleaching”). This unique property of single molecules enables counting the number of molecules involved in a process, and has since been extensively exploited in the field. Measurement of the distribution of fluorescent spot intensity showed a majority of lower intensity spots with a smaller proportion of spots that had approximately double the fluorescence intensity (Figure 2b), indicating the presence of a large number of EGFR monomers and a smaller number of EGFR dimers immediately after the addition of EGF.

One of the main objectives of this work was to test the hypothesis of ligand-induced dimerization for the activation of EGFR. This was achieved by measuring the distribution of intensities of the fluorescent spots at various times after the addition of fluorescent EGF. An increase in the proportion of higher intensity spots would indicate the formation of more dimers, and this was indeed observed (Figure 2c). The increase in dimers was observed before an increase in intracellular Ca^2+^—a secondary messenger dependent on EGFR activation [49]—which occurred after one minute. This supports a model in which EGF binding induces dimerization, which is followed by EGFR activation. Additional information on the EGFR monomer–dimer transition was obtained by simultaneously monitoring the intensity and position of individual EGFR molecules. In some cases, two fluorescent spots came into contact and then moved together, indicating the formation of a dimer from two monomers, each with the ligand already bound. A more common observation was a sudden doubling of intensity of a fluorescent spot, indicating the binding from solution of an EGF molecule to an already existing EGFR dimer with only one EGF initially attached, showing the presence of preformed EGFR dimers.

Sako et al. [46] used two-color SPT to further study the hypothesis that the formation of EGFR dimers complexed with EGF leads to EGFR activation, by investigating the autophosphorylation of EGFR using a monoclonal antibody, mAb74, that only recognizes autophosphorylated EGFR [50]. mAb74 was labeled with Cy3 and introduced into the cytoplasm of the cells, while the extracellular domain of EGFR was labeled with EGF that was tagged with the longer wavelength fluorescent probe Cy5. The co-localization of Cy3 fluorescence with higher intensity Cy5 spots (consistent with the existence of dimers) showed that dimerization is indeed associated with autophosphorylation.

smFRET in EGFR dimers was also pioneered by Sako et al. [46]. EGFRs in A431 cells, which overexpress the receptor [51], were labeled with a mixture of EGF-Cy3 (donor) and EGF-Cy5 (acceptor), and fluorescence intensity was tracked at both wavelengths. Spots showing fluorescence in both channels showed evidence of anticorrelation, with fluctuating levels of FRET efficiency (Figure 3). This was interpreted as being compatible with conformational fluctuations in the dimer. However, only 5% of two-color spots showed detectable FRET, and the authors speculated that the distance between EGF molecules in an EGFR dimer may be greater than measurable by FRET with this donor–acceptor pair. This was an insightful prediction borne out by the crystallographic data published later [47,48]. The distance between EGFs in dimers has subsequently been investigated further with SPT methods, and we return to the topic later in this review.

The two early studies described above established the power of SPT for the characterization of systems such as EGFR in cells. Key molecular characteristics such as motion, association and dissociation, and conformation were all measured using single-molecule techniques. In the 25 years following the earliest publication, SPT-based methods have been used to further elucidate the behavior of EGFR in all these aspects, as well as them being applied to many other molecules. For the remainder of this review, we show how developments in SPT have led us to our current understanding of the structure, dynamics, and function of the EGFR in cells, focusing on four critical areas of EGFR biology.

## 4. Investigating Epidermal Growth Factor Receptor Confinement at the Plasma Membrane: Lipid Rafts and F-Actin

Receptor proteins are abundant in the plasma membrane of mammalian cells. Their role is to detect signals in the extracellular milieu and transduce them across the plasma membrane to initiate signaling networks (see e.g., [52]). Signaling across the plasma membrane proceeds through a subtle balance between negative and positive feedback loops over different scales of time and space that can be easily masked by ensemble averages. Furthermore, the plasma membrane of live cells is itself complex and highly heterogeneous [53], and it is also a highly dynamic barrier [54]. For these reasons, an understanding of the mechanisms regulating EGFR signaling at the plasma membrane requires dynamic investigations of individual receptor behavior and in live cells, as demonstrated by the early seminal work of Kusumi et al. [40]. In this context, SPT becomes a method of choice as it can explore the underlying dynamic mechanisms and allow, as described below, in-depth investigations of the rich, context-dependent signaling responses regulated by the plasma membrane [55].

A key element of plasma membrane organization is said to be the lipid raft, which is a subdomain of the plasma membrane that contains high concentrations of cholesterol and glycosphingolipids [56]. Proteins involved in cell signaling are believed to partition into lipid rafts, suggesting a possible role for rafts in the regulation of signal transduction [57,58]. It has also been suggested that there is a link between lipid rafts and the actin cytoskeleton [59]. The very existence of lipid rafts is still the subject of some controversy [60], but there have been for some time suggestions that EGFR is associated with lipid rafts, and SPT has been used to investigate this. An obvious approach is to track the motion of receptors using SPT, both in the presence and absence of treatments that are said to disrupt the formation of rafts. In 2005, Orr et al. used this approach [61], building on the SPT methods used in the early publications. EGFR and its fellow receptor HER2 were tagged with monoclonal antibodies labeled with the fluorescent probes Alexa 546 and Alexa 647, respectively. The key to investigating lipid raft confinement was a very high precision of single-molecule localization, as raft sizes are said to range from <20 nm to a maximum of around 100 nm [62].

We have described above how the location of single-point emitters such as fluorophores can be determined to be significantly better than optical resolution, by approximating the fluorescence signature (the point-spread function, PSF) with a mathematical function, which is usually a Gaussian [63]. Using this approach, Orr et al. achieved a localization precision of 20 nm. The position of the labeled receptors was recorded with time, and their motion was analyzed through the calculation of MSD plots.

The roles of both F-actin and cholesterol were investigated by tracking receptor motion after the addition of drugs such as cytochalasin D or methyl-β-cyclodextrin, which cause actin depolymerization or cholesterol depletion at the plasma membrane, respectively [64,65]. F-actin depolymerization was found to increase EGFR mobility and increase the size of the restriction domains of both receptors, while cholesterol depletion led to an almost complete confinement of the receptors within nanodomains. Conversely, enrichment of the membrane with cholesterol extended the boundaries of the restricted areas. F-actin depolymerization in cholesterol-depleted membranes partially restored receptor mobility. The authors drew the conclusion that membrane cholesterol provides a dynamic environment that facilitates the free motion of EGFR and HER2, possibly by modulating the dynamic state of F-actin.

For both EGFR and HER2, MSD plots of a similar shape to that shown in Figure 1f were obtained, indicating the restricted motion of the receptor, with EGFR being significantly less mobile than HER2. The receptors were shown to be confined to the microscale regions of the plasma membrane, but also displayed shorter episodes of confinement in the nanoscale regions.

One issue that potentially limits the potential of SPT to investigate EGFR/plasma membrane interactions and/or determine diffusive modes (Figure 1) is the limited duration of fluorescence from organic dyes and fluorescent proteins. Quantum dots (QD) are small (several nm) semiconductor particles that emit light when they are illuminated (see e.g., [66]). QDs are much brighter and, crucially, much more photostable than conventional fluorescent probes, and these properties have been exploited for SPT. The first use of QDs for SPT the EGFR was published by Lidke et al. in 2005 [67], in which EGF was conjugated with QDs and used to label and track EGFR in a number of cell types. A key concept in this study is that with short-lived fluorescent probes, it is sometimes not possible to distinguish between random diffusion and directed transport, because only the initial part of the MSD curve can be captured. This is illustrated in Figure 4, where the data collected for the typical fluorescence duration of a conventional probe (up to 10 s) can be fitted to a straight line, indicating random diffusion. If data are collected for longer, which is possible if QDs are used, an upward curve is revealed, which is typical of directed motion. Taking advantage of the long duration of fluorescence from QDs, the authors identified the retrograde (directional) transport of EGFR on filopodia in A431 cells. The depolymerization of F-actin using cytochalasin D disrupted retrograde transport, whereas the disruption of microtubules did not. In addition, when the concentration of EGF-QD was reduced to very low levels (five pM), only allowing the formation of single EGF-QD-EGFR complexes, the directional motion disappeared, indicating that EGFR oligomerization is required for the active transport of the receptor.

The study by Lidke et al. [67] illustrated how the judicious selection of fluorescent probes, such as QDs, can help with extracting new information using SPT. However, the SPT technique can also be improved in other ways. Underlying any SPT experiment is the computational method that is used to locate and track the fluorescent particles. A major limitation here is the inability of the early SPT methods to function adequately when the density of particles to be tracked is high, which can be often the case. The challenge is to identify individual particle tracks, which becomes difficult as the particles become closer together and their paths sometimes cross. Although tracking at high particle density is challenging, the potential reward is high, as it would allow the acquisition of much more data under a much wider range of conditions. A publication by Serge et al. from 2008 described a solution for SPT at high particle densities, which was referred to as multiple-target tracing (MTT) [30]. Conventional approaches firstly detect the position of particles, and then join together a series of individual locations to provide single-particle tracks. The MTT approach instead links detection and connection with analyses alternately performed using information obtained at both stages using maximum likelihood methods. Combining localization and detection with the integrated historical information of trajectories allows efficient multiple-target reconnection. In addition, a technique known as deflation allows the subtraction of already detected intensities, revealing additional lower intensities that were masked by the higher ones. The MTT method was used to track EGFR labeled with QD-EGF, obtaining impressive maps of EGFR dynamics throughout the cell, which is a process termed “cartography”. This approach built on the previous work by Orr et al. [61] to provide further evidence of the EGFR confinement in sub-micron domains at the plasma membrane, and showed its transient nature. Indeed, the whole cell mapping enabled by MTT showed a complex dynamic picture, with a broad distribution of confinement strengths.

An entirely different single-molecule approach has also been used to investigate the interactions between the EGFR and the plasma membrane. The principle of FLImP is illustrated in Figure 5.

By providing a distribution of the separations of molecules within a diffraction-limited spot, FLImP allows the characterization of dimers and higher-order complexes that cannot be measured by FRET because the distances are too long. Webb et al. [68] used FLImP to show the presence of EGFR dimers in cells, with ~11 nm of intermolecular separation. The presence of higher-order oligomers was also detected. As in previous SPT work, the drugs methyl-β-cyclodextrin and latrunculin were used to disrupt membrane cholesterol and F-actin, respectively, to test their effects on receptor oligomerization. In support of previous work, both cholesterol depletion and F-actin disruption increased the proportion of EGFR oligomers detected by FLImP, supporting the hypotheses that the removal of cholesterol from the plasma membrane activates the receptor by increasing oligomerization, and that cholesterol enhances receptor activation via the modulation of the formation of F-actin filaments.

The picture that emerged from this work is one of complex and dynamic interactions between the receptor and its plasma membrane environment, in which multiple and transient interactions mediated by lipids and F-actin deploy the receptor to critical areas on the cell surface and maintain it as confined under the regulation of a delicate balance of positive and negative feedback loops. We discussed below how these interactions are also critically involved in receptor dimerization, which is the activation switch of EGFR signaling.

## 5. Correlating Epidermal Growth Factor Receptor Dimerization with Plasma Membrane Interactions

Early work showed that the EGFR’s intrinsic intracellular protein–tyrosine kinase activity is stimulated by receptor dimerization [11,69], which results in the autophosphorylation of the receptor’s C-terminal domain in tyrosine residues [70]. Structures of the entire receptor monomer and dimer are not yet available, but a detailed view of the activation mechanism has been developed from studies of the structures of receptor fragments [71]. The EGFR monomer consists of an N-terminal ligand-binding extracellular module (ECM) linked to an intracellular module (ICM) by a single-pass transmembrane (TM) helix. The ECM consists of four domains (DI–DIV) [72]. The ICM includes a short juxtamembrane (JM) segment, a tyrosine kinase domain (TKD), and a disordered carboxy-terminal region, which is where the key tyrosine phosphorylation sites are located (Tyr992, Tyr1045, Tyr1068, Tyr1086, and Tyr1173) [73,74]. The extended conformation of the ECM is stabilized by ligand binding, promoting the formation of back-to-back dimers. In this structure, the ligand does not participate in the binding interface [47,48]. After dimerization, the receptor signals across the membrane. The signal is affected by an asymmetric TKD (aTKD) dimer, through an allosteric interaction between an activator and receiver kinase [75].

Ligand-free, inactive EGFRs have for a long time been believed to be monomers at the plasma membrane, and thought to adopt a tethered conformation via DII–DIV interaction that prevents the formation of a back-to-back dimer [76]. However, over a number of years, evidence has built up for the presence of ligand-free EGFR dimers and possibly oligomers (see e.g., refs. [77,78,79,80,81]). However, it is unclear how signaling from unliganded non-monomers is prevented. One suggestion is that the receptor adopts an inactive symmetric TKD (sTKD) dimer. Such an arrangement has been shown by structures of mutant EGFRs, which have V924R (or V948R) and I682Q mutations at the C-lobe and N-lobe. These mutations inhibit the formation of an aTKD (PDB ID 3GT8 [82], 2GS7 [75], and 5CNN [73]). It has also been proposed that the sTKD might be associated with a side-to-side ECM tethered dimer [83].

Another suggestion is that the sTKD dimer is coupled to an unliganded back-to-back dimer via a C-crossing TM domain dimer. This was proposed as a result of Molecular Dynamics (MD) simulations [84], and would be analogous to the structure of the Drosophila ECM dimer determined by X-ray crystallography [85]. Another model is based on small-angle X-ray scattering data from the Caenorhabditis elegans EGFR [79]. In this case, a ligand-free back-to-back dimer that resembles the ligand-bound dimer is formed, requiring autoinhibition by only the sTKD dimer [86]. Electron microscopy (EM) revealed other possibilities. Images of the purified EGFR mutant Δ998-EGFR [87,88] resulted in the proposal of a “stalk-to-stalk” dimer mediated by type I tyrosine kinase inhibitors (TKIs) that bind at the ATP-binding pocket in a reversible manner and inhibit phosphorylation at the C-terminus. This could promote kinase-mediated interactions in the proposed dimer [87]. The key to understanding these possible autoinhibition mechanisms has been the development of techniques that allow the measurement of structure on cells at high resolution.

SPT has played a fundamental role in characterizing the properties and function of ligand-bound and ligand-free EGFR dimers in cells, and has been used to investigate the interactions between EGFR dimers and EGF. In 2006 Teramura et al. reported a study in which EGF was conjugated with the fluorescent probe rhodamine, and used to label EGFR in HeLa cells [89]. EGF binding to the receptor was shown by the detection of a fluorescent spot, with the binding of a second EGF resulting in a step increase in the fluorescence intensity of the spot. They analyzed the kinetics of EGF binding, and found that singly-labeled EGFR dimers bind a second EGF molecule with a much higher affinity than binding to a monomeric binding site. This observation indicated positive EGF binding cooperativity in the EGFR dimer, which was in contrast to most of the reports in the literature, in which negative cooperativity was instead detected (e.g., [90]). To explain this discrepancy, Teramura et al. proposed a model based on the crystal structures of the extracellular domain [48] and the tethered monomer [76], in which the binding of a single EGF results in the formation of an extended intermediate, which is “primed” to bind the second EGF without the need for a large conformational change from tethered to extended (Figure 6). One potential uncertainty resulting from the used approach is that because of resolution limitations, it was not possible to determine with certainty that two bound fluorescent EGF molecules are attached to a dimer or two receptors that are separate but somehow linked together (e.g., co-confined).

As shown by the pioneering FRET work of Sako et al. [46], SPT experiments can also be used to obtain information on receptor responses to ligands that complement that from ligand-binding kinetics. Both wavelength and polarization information can simultaneously be obtained, with the potential of quantitatively measuring changes in intramolecular distances and angles. Such an approach, which is named “multidimensional single molecule imaging”, was used to investigate changes in EGFR conformation [21]. In this experiment, Webb et al. used the microscope to collect four spatially identical images, which differed solely in their spectral and polarization properties. EGFR in A431 cells were labeled with EGF tagged with the Cy3 and Cy5 fluorophores, which was the same FRET pair previously used by Sako et al. [46]. The intensity of fluorescent spots was recorded with time at both FRET donor and acceptor emission wavelengths, and at polarizations that were parallel and perpendicular to the fluorescence excitation, producing traces such as the ones shown in Figure 7. In this example, at the start, there is energy transfer from Cy3 to Cy5 with a FRET efficiency of 0.8, so the acceptor emission is predominant. At point X, there is a rise in donor fluorescence without any significant change in acceptor fluorescence (Figure 7a). This would be consistent with a second EGF-Cy3 (donor) molecule joining the complex, resulting in a drop in FRET efficiency (Figure 7b). At point Y, there is a smaller drop in FRET efficiency, which corresponds to a change in the polarization composition of the fluorescence intensity. At this point, the parallel polarization traces are anticorrelated, while the perpendicular acceptor trace is unaffected, so the FRET change was attributed to a change in the relative angle between the donor emission and acceptor excitation dipoles, indicating a conformational change in the EGFR.

However, the publication of the “back-to-back” EGFR dimer structure in 2002 [47,48] had already raised a crucial question with respect to the FRET studies of the receptor. This dimer structure would result in a separation of ~11 nm between the EGF molecules bound to the EGFR (larger if the size of the probe is considered). This separation is longer than the range over which FRET occurs, yet FRET is still detected in measurements of liganded EGFR on cells (see e.g. [91]). A single-molecule FRET study from 2008 [92] demonstrated two FRET states, suggesting two interfaces between the EGFR ectodomains, in which neither were compatible with the crystallographic dimer. One interface would place the EGF molecules very close together (<5.5 nm). A speculative model was proposed in which these short distances arose from the formation of oligomers. The authors used a non-single molecule FRET method to measure the distance from the EGF to the plasma membrane, and showed that the EGF in the proposed oligomer structure was closer to the membrane than would be expected if the receptors were standing upright in the back-to-back dimer. The model that was proposed, based on a crystallographic head-to-head contact in the back-to-back structure [47], is shown in Figure 8, and the authors speculated that the different extracellular domain configuration could result in a different intracellular kinase domain arrangement, providing a means for the EGFR to achieve multiple levels of signaling. The potential importance of EGFR oligomerization is being increasingly recognized, and has since been investigated using multiple SPT techniques; we return to this topic in the next section of the review.

Given that the dimensions of the EGFR dimer were too large for FRET, meaning that FRET was not a reliable probe of receptor dimerization, the mechanisms of EGFR dimerization have also been investigated using alternative one and two-color SPT techniques. In the first study using SPT to investigate the interdependence between diffusion dynamics and the stoichiometry of interacting species, Chung et al. tracked EGFR in Chinese hamster ovary (CHO) cells by labeling the receptor with anti-EGFR FAb fragments linked to one-color quantum dots [93]. This paper provides a good demonstration of how diffusion rates measured from SPT can be used indirectly to infer the size of labeled complexes. They measured the size of complexes by measuring their rates of motion in the membrane, after confirming by tests in model membranes that the diffusivity follows the Stokes–Einstein relationship with the diffusion rate proportional to 1/R, in which R is the radius of the protein. Therefore, fast and slow diffusion rates were interpreted as corresponding to monomeric and dimeric EGFRs, respectively. Using this method with a non-ligand label, they observed for the first time the spontaneous formation of ligand-free dimers that were primed for ligand binding and signaling, as suggested by other studies [94,95]. Unliganded EGFR was shown to fluctuate continuously between the monomeric and dimeric state. Structure-based models had previously suggested that receptor dimerization results from a ligand-induced conformational change in the ectodomain that exposes a loop (dimerization arm) that is required for receptor association [47,48]. This hypothesis was tested by tracking EGFR in which the dimerization arm had been deleted. These experiments showed that the dimerization arm is not required for dimerization, but that it did stabilize both ligand-free and EGF-induced dimers [93]. Further dynamic SPT studies have examined how oligomers might be formed from different dimer species, which were identified as predimers (no ligand bound), heterodimers (one EGF bound), and homodimers (two EGF bound, termed “signaling dimers”) [96]. This report identified oligomers consisting of up to 8 EGFRs induced by EGF, and demonstrated directed motion of EGFR, suggesting that EGFR activation is propagated along predimers bound to the cytoskeleton. A dynamic system involving switching between heterodimers and homodimers was suggested as a mechanism for EGFR activation and cell signal amplification.

Meanwhile, as shown by Chung et al. [93], one-color SPT can reveal much about the dynamics of receptor dimerization; for a more complete understanding of the dimerization mechanism and its kinetics, it is important to directly visualize the dimerization process in two colors and link it to cell signaling. A combination of two-color SPT, which is a more advanced analysis, and the use of a tyrosine kinase inhibitor provided additional information on the dissociation rates of different EGFR dimerization states and the link to signaling [77]. Using bright QDs also allowed a localization precision of 40–60 nm, and using a hidden Markov model for the first time [97]. This method generates a maximum likelihood estimate of the kinetic rate constants for transitions between states. The dissociation rates were elegantly derived from the durations of the correlated motions of pairs of EGFR receptors in which their separation was continuously measured. This approach can be used to identify hidden states that reflect the underlying behaviors of the proteins. Following the common theme in the field, in this case, a three-state model was required to fit the data: ligand-free, transient co-confined, and ligand-bound. This important work showed that the stability of the dimers is governed by ligand occupancy; EGFRs having two bound ligands were longer-lived, with a k_off_ independent of kinase activity. Using a TKI inhibitor showed that changes in EGFR diffusion were linked to the receptor’s activation status. As in previous studies described above, the authors reported the transient co-confinement of receptors, promoting dimerization, while blocking kinase activity or disrupting actin networks resulted in the faster diffusion of the dimers. These results implicate both signal propagation and the cortical cytoskeleton in the reduced mobility of signaling-competent EGFR dimers.

More recent SPT studies have attempted to further understand how interactions with plasma membrane lipids and lateral receptor confinement may regulate EGFR dimerization. In 2015, Lin et al. used SPT to investigate how ligand binding and the dimerization of EGFR are interdependent with EGFR interactions with the plasma membrane, again using drug treatments to disrupt the lipid raft domains [98]. In addition to determining the extent of receptor confinement using single EGFR particle tracks and MSDs, the authors also used dual-color SPT to investigate the correlated motion of pairs of EGFR molecules, showing dimer formation with pairs of EGFR undergoing correlated motion for 10 to 30 seconds. Results were interpreted in the light of simulations based on an energetic model of compartmentalized receptor diffusion at the plasma membrane; the main findings of this work were that after ligand binding, EGFR molecules can relocate into nanodomains, whereas unliganded species remain outside these cholesterol-enriched domains. Additionally, lipid nanodomains surrounding two liganded EGFRs were shown to merge during their correlated motion, and that the transition rates between different diffusions states of liganded EGFRs are regulated by the lipid domains. Further studies by the same group [99] found that unliganded EGFRs may reside close to cholesterol-rich membrane regions, and move into them on ligand binding. The authors also demonstrated an association between cholesterol and the stability of correlated motion of activated EGFR dimers, and concluded that cholesterol plays a role in the ligand-induced dimerization of the receptor.

One of the reasons that the EGFR has been the subject of such intensive investigation is its role in the formation and growth of tumors. EGFR mutations have been implicated in a number of human cancers, and anti-cancer drugs have been developed that aim to block EGFR signaling. SPT techniques have been used to study how both mutations and drugs affect the dimerization of EGFR in cells. Gefitinib is one of a number of small-molecule TKIs that attempt to block signaling by competing with ATP binding [100]. SPT experiments have shown that gefitinib, which blocks tyrosine phosphorylation, stabilizes the EGFR ligand-bound homodimer, confirming previous data that shows that the phosphorylation state of EGFR is involved in modulating dimer stability. Expanding on their previous two-color single particle work [77], the dimerization of EGFR mutants associated with lung cancer was also studied with SPT by Valley et al. [101], showing that mutations associated with non-small cell lung cancer result in the formation of stable, ligand-independent dimers. This paper provides a good illustration of how the combination of SPT with other techniques can reveal additional information. The ligand-independent aggregation of EGFR mutants was confirmed using two-color super-resolution localization microscopy, while FRET was used to investigate receptor conformation. The latter showed that the L858R kinase mutation (using a numbering that includes a 24 aa signaling peptide) alters ectodomain structure such that unliganded mutant EGFR adopts an extended, dimerization-competent conformation, while the mutation of the putative dimerization arm confirmed a critical role for ectodomain engagement in ligand-independent signaling. From the combined data, the authors proposed a model in which the dysregulated activity of the mutants is driven by coordinated interactions involving both the kinase and extracellular domains, leading to enhanced dimerization.

The series of studies described above, which were made possible by increasingly sophisticated SPT methods, provided significant insights into how EGFR dimerizes, how the dimerization process is related to other cell components such as the cytoskeleton and the plasma membrane, and how dimerization is related to cell signaling in the wild-type and mutant receptors. They have also provided evidence that EGFR exists in the cell as larger oligomers. Researchers have also turned to SPT to further investigate the nature and role of these oligomers, and this is reviewed in the next section.

## 6. Investigating the Existence and Function of Epidermal Growth Factor Receptor Oligomers and Clusters

Although the mainstream of EGFR research has largely focused on a dimerization-dependent activation mechanism, pioneering work using FRET and image correlation spectroscopy suggested that the formation of tetramers also plays a crucial role in EGFR signaling [78]. Several subsequent studies suggested that EGFR also exist in the form of higher-order oligomers or large clusters containing on the order of 10^1^–10^3^ proteins [102,103,104]. The local density of receptors and/or the composition of the local plasma membrane also appears to regulate the size of the clusters [80,105]. In addition, distinct EGFR cluster populations may co-localize to caveolae, lipid raft domains, clathrin-coated pits, and/or unstructured plasma membrane regions [103,106], relating yet again the oligomeric state of the receptor with its interactions with the plasma membrane and associated proteins. It is possible that these plasma membrane domains might dynamically sequester EGFR to promote clustering or dissociation in order to regulate its function [107,108]. However, despite the potential importance of oligomerization in EGFR signaling, key aspects of EGFR oligomers remain unclear, including their nature and activation status, and how interactions with the plasma membrane contribute to the oligomerization process.

SPT has been used to elucidate some of these important questions. An early study by Ichinose et al. in 2004 took advantage of the phenomenon of single-step photobleaching to count the number of activated EGFR in clusters [109]. Live A431 cells were incubated with rhodamine-EGF for time periods ranging from one minute to 30 minutes. The cells were then fixed and chemically permeabilized to allow the entry of Alexa488-labeled Fab’ fragments of a monoclonal antibody, mAb74, against an activated form of EGFR. This antibody recognizes a conformational change in the cytoplasmic domain of EGFR after autophosphorylation [50]. The co-localization of rhodamine and Alexa 488 fluorescence was interpreted as reporting activated EGFR. The number of EGF-bound and activated receptors per cluster were determined by counting the number of photobleaching steps in tracks of fluorescence intensity versus time. Longer incubation times resulted in higher numbers of phosphorylated EGFR per spot, suggesting the growth of activated clusters. The number of phosphorylated receptors was found to be three times larger than the number of EGF molecules, indicating that the EGF signal was amplified in the EGFR activation process. The fact that there was not a simple linear correlation between EGF binding and EGFR activation could not easily be explained based on models that were available at the time, but further light was shed on this by later SPT studies, as described below. However, the authors were able to conclude that clusters of activated EGFR form as a consequence of the aggregation of unliganded EGFR onto ligand-bound activated EGFR, and that the dynamic clustering of EGFR was the source of amplification of the EGF signal. Additional light was shed on the amplification process by a later SPT study, which demonstrated that the activation of EGFR is supported by clustering in clathrin-coated pits, and amplified by cross-phosphorylation [110]. Other SPT studies also showed a complex picture, with multiple EGFR cluster states and intermediates showing multistate temporal and spatial dynamics [111,112].

The evidence for the existence of higher-order oligomers and clusters of EGFR continued to accumulate in the years following the early SPT studies through methods such as near-field scanning optical microscopy (NSOM) [103], number and brightness [113], and electron microscopy (EM) [102]. However, their functional relevance remained the subject of considerable controversy. The first study using SPT to measure the dynamics of EGFR clusters in living cells was published by Boggara et al. in 2012 [114]. As a first step, they focused on the lateral diffusion of clusters on the apical surface of live cells. Brightness analysis was used to estimate that there were around 50 receptors per cluster. The diffusivity of these EGF-bound clusters was similar to that of a ligand-bound EGFR dimer (using previously published values for the latter). As previously shown for monomers and dimers, MSD plots for the EGFR clusters also showed simple Brownian motion, directed motion, and confined motion. In the same study, the influence of the cytoskeleton was investigated by disrupting both actin and microtubules. Actin disruption increased the diffusivity of the clusters, while the disruption of microtubules significantly reduced mobility. The influence of microtubules was found to be greater for larger EGFR oligomers or clusters, supporting the “oligomer-induced trapping model” proposed from early single-molecule studies [115].

The evidence for the existence of higher-order oligomers, and for a role for them in EGFR activation, was greatly strengthened by the studies described above. However, the structure and exact function of those oligomers remained elusive. Two studies published in 2016 showed that the combination of SPT methods with other techniques, including MD simulations, could provide valuable information on the nature of the oligomers. One important study by the Kuriyan lab surveyed a number of EGFR mutations in an attempt to understand the nature of the interfaces involved in oligomerization [116]. EGFR oligomers were detected by counting photobleaching steps in single-molecule images. The cells used (*Xenopus* oocytes) had very low EGFR densities (one to five molecules per μm^2^), and in the absence of EGF, the receptor was found to be predominantly monomeric in these cells. The addition of EGF promoted the formation of dimers and higher-order oligomers, with around 50% of the spots showing multistep photobleaching after addition of the ligand. Multimerization could be blocked by mutations in a specific region of domain IV of the receptor, comprising residues between Val 526 and Val 592 in the EGFR ectodomain. The mutations were shown to reduce the autophosphorylation of the C-terminal tail of EGFR and attenuate the phosphorylation of PI3 kinase, which is recruited to EGFR. The authors also showed that allosteric activation is restricted to dimers, i.e., dependent, as expected, on the formation of the aTKD dimer [117], in which sites in the tail that are proximal to the kinase domain are phosphorylated in only one subunit. This work also showed that multimerization is necessary for the robust phosphorylation of phosphatidylinositide 3-kinase (PI3K), and EGFR signaling effector protein, which is recruited to the proximal part of the tail. The SPT data were used to impose constraints on MD simulations, allowing the production of a model for EGFR oligomerization (Figure 9), based on the self-association of ligand-bound dimers. In this model, the majority of the kinase domains are activated cooperatively, boosting tail phosphorylation.

The combination of SPT and MD simulation was also used by another group to investigate the structure and function of oligomers. In this case, the FLImP method was used to map out intermolecular distances within EGFR complexes with an unprecedented 4.8-nm resolution, and the data was used to constrain long-timescale MD simulations [39]. The authors showed that a fundamental requirement for the effective phosphorylation of the receptor is the formation of at least tetramers. Although ligand-induced dimerization is still fundamental to the activation process, the formation of oligomers does appear to be a requirement to allow substrates to access the activated kinases. A structural model that is somewhat different from the one described above was proposed for EGFR oligomers (Figure 10). In this model, two EGF molecules are bound to an oligomer, which is formed by two neighboring EGFR molecules, using a similar interaction to that used for EGF binding. Measurements of EGFRs with mutations in the intracellular domain showed that the interactions of this domain are also essential for the formation of oligomers.

The suggestion from this study is that EGFR signaling is biphasic, and that this is controlled by the balance of receptor dimers and oligomers, which are in turn determined by the concentration of the ligand and the binding affinity of the receptor. In the model, a high level of receptor autophosphorylation can be achieved with a low ligand concentration, and this can explain the ‘superstoichiometric’ signaling behavior displayed by EGFR at the low EGF concentrations we discussed earlier [109]. In this model, Grb2:EGFR recruitment stoichiometries can be as high as 3.5:1 [119].

The combination of FLImP and long-timescale MD simulations also proved to be powerful in understanding the ligand-free complexes of EGFR formed in the basal state, and crucially in understanding the autoinhibition mechanisms that prevent the activation of unliganded EGFR dimers and higher-order oligomers. A study by Zanetti-Domingues et al. [120] demonstrated the existence of chains of receptors of varying lengths through using FLImP combined with FRET in fixed cells and SPT in live cells, which were assembled through a head-to-head interaction of the extracellular domain. The arrangement that results from this head-to-head interaction prevents kinase-mediated dimerization. This study has important implications in understanding the role of mutations in EGFR dysregulation. The autoinhibited head-to-head chains are broken by mutations or intracellular treatments that allow dimerization, resulting in the formation of flexible and unextended stalk-to-stalk dimers. Coupling across the membrane occurs between these dimers and active asymmetric tyrosine kinase dimers, and extended dimers coupled to inactive symmetric kinase dimers. From these data, it appears that dysregulation involves populations of symmetric and asymmetric kinase dimers, co-existing in equilibrium, and modulated by the C-terminal domain. A cartoon of how the different ligand-free species might co-exist in the plasma membrane is shown in Figure 11.

The studies described above show clearly how SPT techniques have not only confirmed the existence of EGFR oligomers and clusters in both the basal and activated states, but also provided detailed information on the structure of the oligomers; they also indicate how this is related to function both in the normal and dysregulated receptors. The power of combining data from SPT with MD simulations is of particular note. This combination can work as a “virtuous circle”, with simulations suggesting possible models that can be confirmed or refuted using SPT, and SPT providing structural and dynamic constraints that can be used to build better models.

## 7. Interactions with Other Receptors, the Endocytic Pathway, and Signaling Effectors

The phosphorylation of the tyrosine residues of the EGFR C-terminal tail upon the binding of EGF triggers a string of responses leading to modified cellular behavior that is affected through a series of secondary events, which are typically initiated through the direct docking of effectors and scaffolds (with associated effectors) at appropriate phosphotyrosine residues. The pattern of phosphorylated tyrosine motifs is specific to each member of the EGFR family [121]. Docking is accomplished in a selective fashion to provide particular responses through the abilities of the SH2 (Src Homology 2) and PTB (phosphotyrosine binding) domains [122] (and possibly some C2 domains [123]) that are present in these effectors and scaffolds to recognize phosphotyrosine residues in a sequence-specific manner. For example, EGFR dimers recruit the adaptor Grb2 through tyrosine-based motifs (leading via Sos to the activation of the mitogenic Ras/MAP pathway), Shc, and the transcription factor STAT5, whereas no docking site was found for PI3K [124].

The phosphorylation of the C-terminal tyrosine residues also activates intracellular pathways for signal attenuation (receptor desensitization, down-regulation, endocytosis, and trafficking), amplification processing [125], and cross-talk with other receptors and the plasma membrane [126]. Understanding the nuances of these interdependent processes is paramount to understanding how such interactions at the plasma membrane contribute to growth and invasion. In this final part of the review, we summarize how SPT has been used to investigate how EGFR interacts with other components of the signaling and down-regulation pathways.

We have mentioned above in the ‘Background to SPT Techniques’ section that SPT can be used to investigate the dynamics of molecular interactions in cells, including measuring the strength of interaction by determining on and off rates. The technique has also been used to investigate the binding of EGFR to the adaptor protein Grb2 [127]. In this study, Grb2 was linked to the fluorophore Cy3, and tracks of labeled Grb2 molecules were recorded as they interacted with EGFR bound to EGF in A431 cells [128]. On and off rates for the EGFR–Grb2 interaction were measured, with three states being observed. The major component (>89%) had a dissociation constant of 7.5–8.1 s^−1^, which was between 2–11% of the interactions had a dissociation constant of 1.6–2.6 s^−1^, and a third, minor component had a much slower interaction. These results suggest that EGFR and Grb2 have multiple binding sites with different dissociation rates. Increasing the Grb2 concentration by a factor of 10 resulted in an approximately threefold increase in the reaction frequency. The authors suggested that the results indicate the presence of multiple conformational transitions of the binding site. The hypothesis proposed was that Grb2 repeatedly interacts with EGFR for short times, accelerating substrate transitions in a way that increases the time constant of association. This was proposed as a mechanism by which the cell could compensate for fluctuations in the Grb2 expression level while maintaining stable signal transduction.

Another important EGFR interaction is believed to be with G protein-coupled receptors (GPCRs), which have been reported to be involved in the transactivation of EGFR [129]. Both ligand-dependent and ligand-independent mechanisms of transactivation have been proposed. In the ligand-dependent mechanism, the stimulation of GPCRs via selective agonists triggers the membrane-bound matrix metalloproteases (MMPs)-mediated proteolytic cleavage of a pro-ligand, generating a cleaved ligand EGF that binds to EGFR and induces signal transduction [130,131,132]. The ligand-independent mechanism requires the activation of the intracellular protein Src to directly phosphorylate EGFR, without the involvement of MMP or EGF [133,134,135]. These previous studies have been based largely on biochemical assays, and therefore could not directly reveal the spatial and temporal dynamics of EGFRs on living cells during GPCR-mediated transactivation. Again, the use of SPT has proved to be useful in shedding further light on the interactions of EGFR. A recent study used SPT to track both EGFR and the GPCR beta-2 adrenergic receptor (β2-AR) in live HeLa cells [136], directly observing the stoichiometry and dynamics of EGFR transactivation. They used isoprotenerol (ISO) to activate β2-AR, and found that this induced EGFR dimerization and slower diffusion. This dimerization was prevented by either the knock-down or inhibition of Src, proving a role for Src in EGFR transactivation by GPCRs.

Finally, several studies have also used SPT to investigate EGFR’s place in the endocytic pathway. Xiao et al. used SPT to determine EGFR mobility and confinement through the determination of MSDs in cells, investigating endocytosis by disrupting clathrin-coated pits and caveolae [137]. They found evidence for the occurrence of both clathrin-dependent and caveolae-dependent EGFR endocytosis, with caveolae becoming involved at higher EGF concentrations. Incidentally, this study is also of interest because it was one of the earliest to use EGFP as a fluorescent marker. In principle, this has advantages over probes such as labeled EGF or antibodies, as it should ensure that all of the receptors are visible. The involvement of clathrin-coated pits was also demonstrated in another SPT study that showed that the activation of the receptor was amplified by cross-phosphorylation, which itself was enabled by clustering within the pits [110].

## 8. Conclusions

Using the EGFR as an example, we have shown how SPT techniques have developed over the last few decades, and have been able to provide detailed information on the structure, function, and interactions of this important system in the cellular context. In particular, many of the studies that have been discussed have shown how SPT, because it observes individual molecules rather than an ensemble average, is the only method that is able to untangle the complex multistate dynamics of molecules in cells. In addition, the ability to locate individual molecules with nanometer precision gives structural information in the cell that provides insights into the functioning of molecular complexes. Another common theme is that the combination of SPT information with atomic resolution structures from crystallography and MD simulations enables the building of detailed models of receptor complexes, which in turn increase our knowledge of how these complexes function in health and disease. Due to these advantages, SPT has been used not only to investigate EGFR, but also a large number of other systems, including for example immunoglobulin E receptors [138], CD4 receptors [139], α3-containing glycine receptor [140], and the neurotrophin receptor TrkA [141], among many others. Very recently, Yasui et al. [142] developed an automated single-molecule imaging system in living cells, applying for the first time artificial intelligence algorithms to the imaging and analysis of EGFR, including changes in lateral motility at the plasma membrane and the response to various ligands and drugs. It is expected that the combination of the impressive portfolio of SPT methods described in this review would benefit from being assisted by artificial intelligence (AI), in particular by decreasing the experimental challenge associated with these crucial but often difficult experiments.

There remain a number of areas in which SPT could be improved in order to extract more or better quality data from biological systems. There is to date no ideal fluorescent probe; QDs are relatively large, can be toxic to cells, and are still subject to blinking. Organic fluorophores photobleach, and are therefore unsuitable for monitoring events of relatively long duration. In addition, the problem remains of delivering the fluorophore to the target molecule. Fluorescent ligands activate their target, and are therefore unsuitable for monitoring the basal state. Antibodies and antibody-like molecules always bring with them the possibility of the perturbation of the system. Many probe molecules do not cross the plasma membrane, and are thus unsuitable for labeling, for example, the intracellular domain of receptors. Therefore, probe development remains a significant area of development for SPT work.

Similarly, there is significant scope for the improvement of data analysis. Tracking single molecules that blink and cross paths in a crowded environment in the presence of high levels of background fluorescence is extremely difficult. Generally, tracking algorithms requires the use of assumptions on how the molecules will move. This has the potential to introduce artefacts and bias results. Ideally, the goal is the globally optimal spatiotemporal solution to SPT. This approach tries each possible choice of particle reconnections, associated motion parameter values, and particle states, and compares the consequences of each choice along the entire length of the tracks. Achieving this globally optimal solution has been the goal of SPT for decades, but it has remained computationally prohibitive because of the colossal size of the configuration space of particle reconnection possibilities at the high particle density, low signal-to-noise ratio, and fast particle movement that are typical of single-molecule images in cells. New approaches and the availability of ever-increasing amounts of computing power may put this goal within reach. Recently, neural network-based approaches have been described that have great potential for automating single-particle tracking, significantly reducing the requirement for user intervention, and therefore the possibility of user-introduced bias [143].

For EGFR and other systems, we increasingly expect to see the use of combined SPT methods to determine simultaneously, for example, oligomer size and diffusion properties. Looking at systems such as EGFR in the context of the whole cell requires the development of three-dimensional (3D) tracking methods that can follow processes from the plasma membrane to the nucleus. Three-dimensional SPT methods and instrumentation are beginning to appear (see e.g., [144,145]), but more development is required for them to become routinely usable by life sciences researchers. Ultimately, as these methods develop, we might see SPT extended further to tissues and organisms, opening up new areas of research.

## Figures and Tables

**Figure 1 mps-02-00012-f001:**
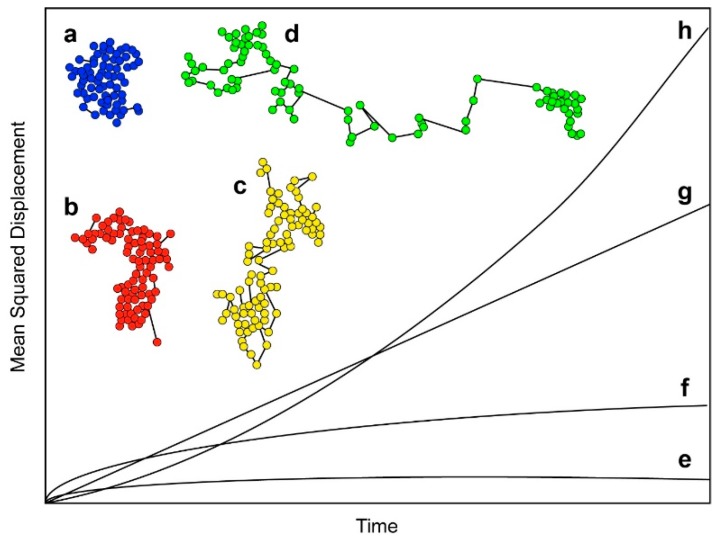
Types of receptor motion measurable by single-particle tracking (SPT) (adapted from Bacher et al., 2004 [45]): (**a**) Confined molecule; (**b**) Restricted motion; (**c**) Simple diffusion mode; and (**d**) Directed motion. (**e**–**f**) show mean squared displacement (MSD) plots that would be obtained for the types of motion shown in (**a**–**d**), respectively.

**Figure 2 mps-02-00012-f002:**
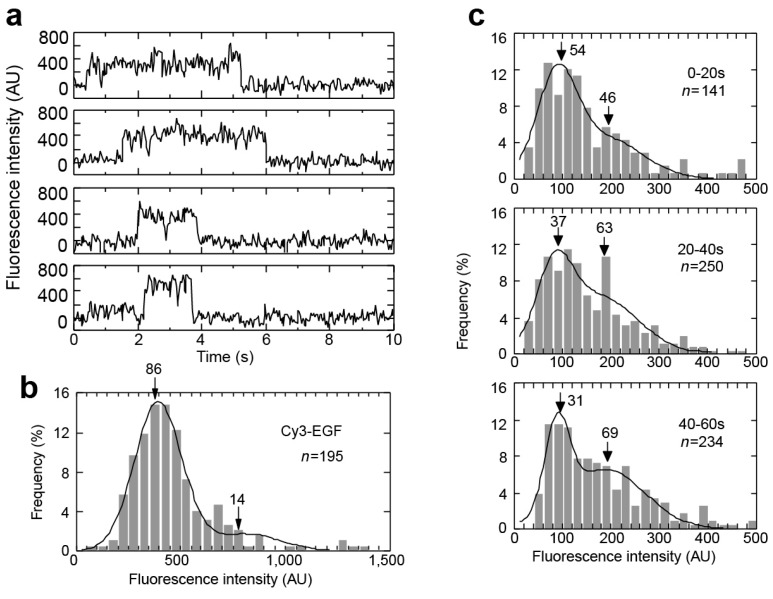
Single-molecule fluorescence data from Sako et al. (2000) [46], from fluorescent epidermal growth factor (EGF) bound to epidermal growth factor receptor (EGFR) in A431 cells: (**a**) Single-molecule fluorescence intensity traces; (**b**) Distribution of fluorescence intensity immediately after the addition of EGF; (**c**) Distribution of fluorescence intensity at time intervals following the addition of EGF.

**Figure 3 mps-02-00012-f003:**
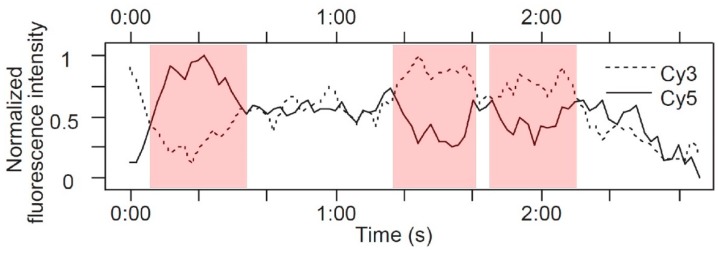
Typical anticorrelated single molecule traces showing the presence of single-molecule Förster resonance energy transfer (smFRET) (adapted from Sako et al. [46]). The presence of FRET is shown by increases in acceptor (Cy5) fluorescence and decreases in donor (Cy3) fluorescence (highlighted in red).

**Figure 4 mps-02-00012-f004:**
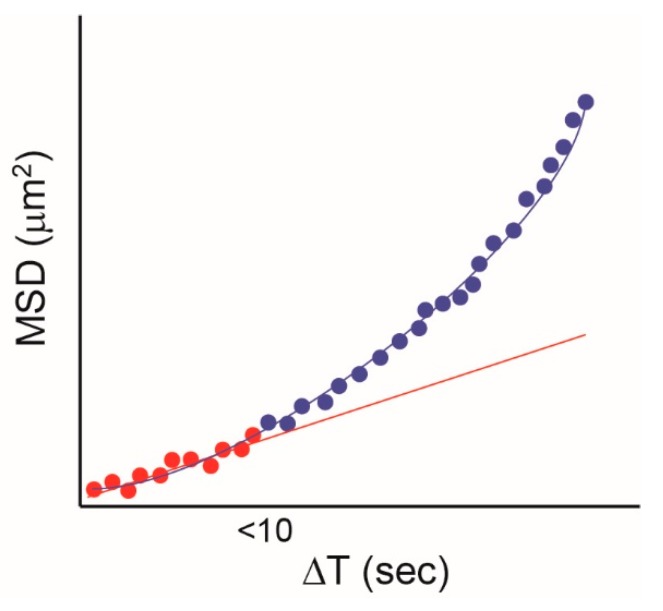
Simulated data showing the potential benefits of using quantum dots for SPT. Data that were only collected for the duration of fluorescence of a conventional fluorescent probe (red dots) showed a straight line fit (simple diffusion), but data that were collected for longer (blue dots) revealed an upward curve (directed motion).

**Figure 5 mps-02-00012-f005:**
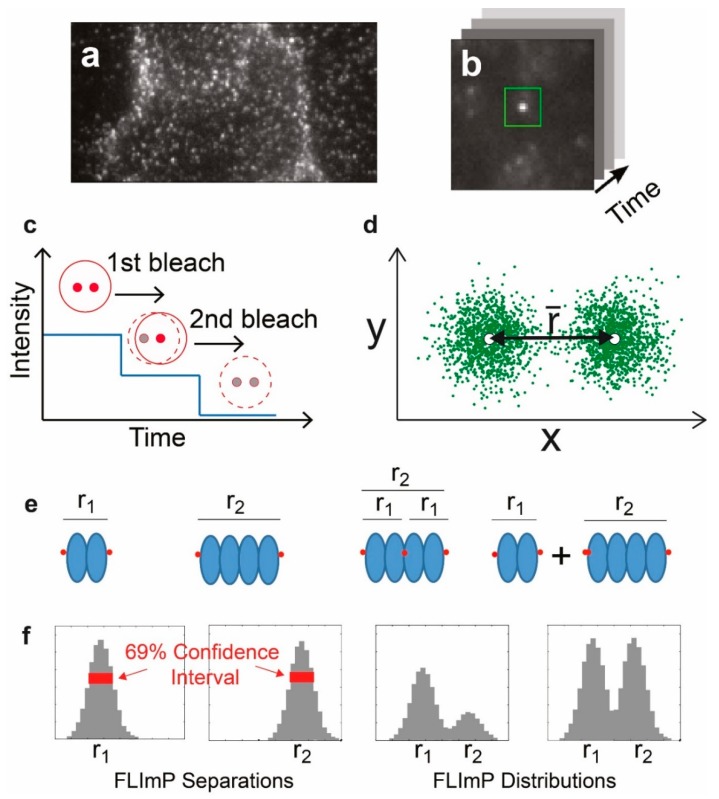
Determination of molecular separations using fluorophore localization imaging with photobleaching (FLImP). (**a**) A total internal reflection fluorescence (TIRF) microscope is used to collect single-molecule images from cells, and fluorescence intensity is measured vs. time for spots from individual complexes (**b**). (**c**) The fluorescence intensity of a spot containing two fluorophores decays in two photobleaching steps, with the centroid position shifting when one fluorophore bleaches. (**d**) The best intensity, *x*–*y* positions, and the full-width at half-maximum of the point spread function for each fluorophore are obtained using a global least squares seven-parameter fit. The fluorophore separation ($r=\;\sqrt{{\left(x_{1}-x_{2}\right)}^{2}+\;{\left(y_{1}-y_{2}\right)}^{2}}$) is calculated from the fit, with the precision being determined by the localization error. (**e**) Examples of FLImP distributions that might be measured in EGF (red) bound to EGFR, showing a two-ligand dimer and tetramer, a three-ligand tetramer, and a dimer/tetramer mixture. (**f**) Simulations of empirical posterior distributions (or FLImP measurements) for pairwise ligand separations from each example system. Confidence intervals of 69% are highlighted. Figure adapted from Needham et al. 2016 [39].

**Figure 6 mps-02-00012-f006:**
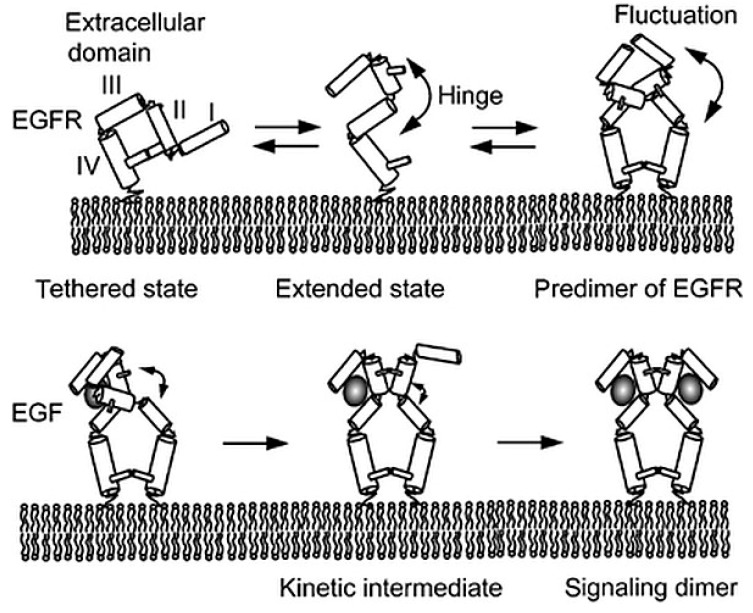
Model for the formation of signaling dimer EGF/EGFR complexes (figure from Teramura et al. [89]). The EGFR fluctuates between the tethered and extended states, and can form predimers that stabilize the extended state. The binding of a single EGF to a predimer results in a conformational change, increasing the affinity for a second EGF.

**Figure 7 mps-02-00012-f007:**
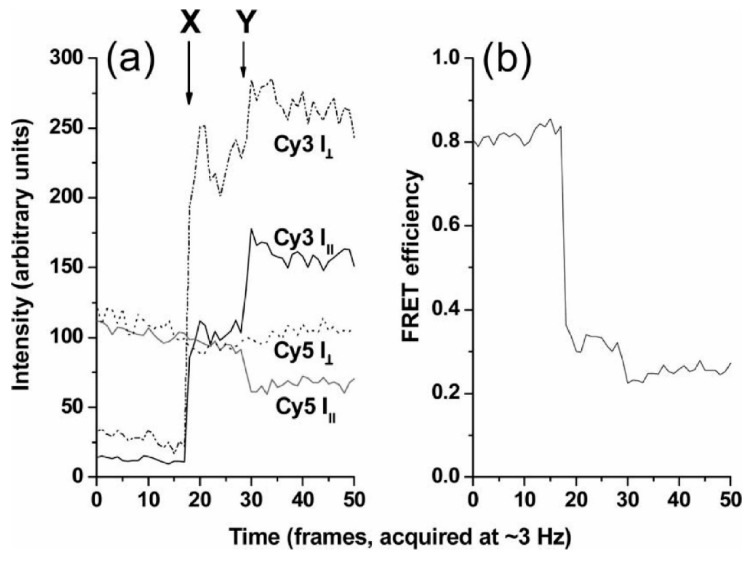
Polarization-resolved single-pair FRET in live cells, taken from Webb et al. [91]. (**a**) Temporal variation of polarization and wavelength-resolved intensity from EGF–Cy3 and EGF–Cy5 molecules bound to EGFR in A431 cells; (**b**) Corresponding variation in FRET efficiency.

**Figure 8 mps-02-00012-f008:**
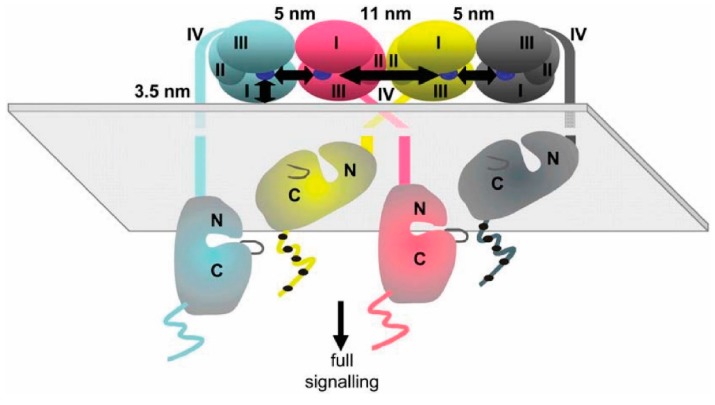
Model of EGFR signaling complex derived from single-molecule and ensemble FRET measurements described in Webb et al. (2008) [92]. EGF is bound to receptors in a tetramer involving a combination of two distinct configurations. Cyan, magenta, yellow, and grey structures each represent an individual EGFR molecule in the complex. The Roman numerals I–IV refer to domains of the EGFR, and “N” and “C” indicate the positions of the N and C termini of the intracellular domain of the receptor.

**Figure 9 mps-02-00012-f009:**
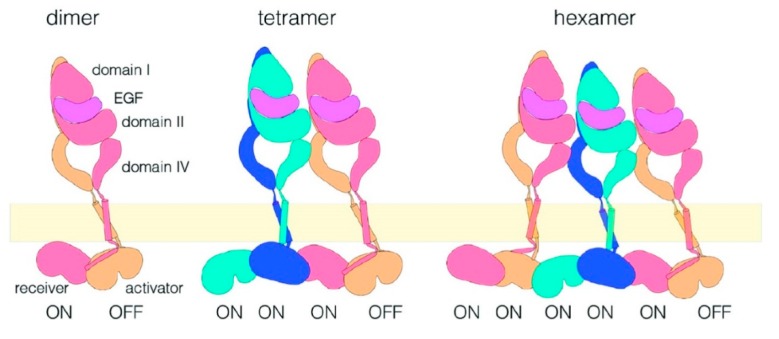
Model for EGFR oligomerization (figure adapted from Huang et al. (2016) [116]). The model is built from the known dimeric arrangements of the extracellular module, the transmembrane helices, and the kinase domains, and these have been connected to produce a model for dimeric full-length EGFR, as described [84,118]. The model for EGFR tetramers was obtained by packing the dimers against each other as rigid bodies. Repeating the interactions allows the building of higher-order oligomers such as tetramers.

**Figure 10 mps-02-00012-f010:**
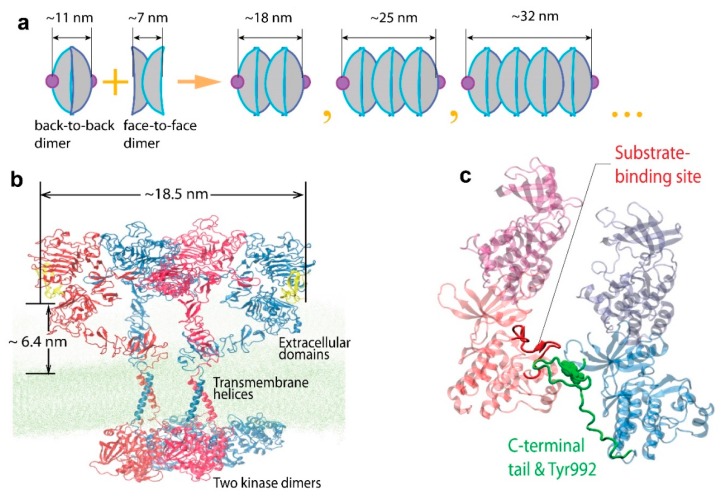
Model for EGFR oligomerization (figure adapted from Needham et al. (2016) [39]). (**a**) Illustration of a proposed oligomerization scheme for the EGFR extracellular domain, which depends on repeating back-to-back and face-to-face interactions of the receptor. (**b**) A complete model of an EGFR tetramer structure, which was formed by the dimerization through the face-to-face interaction of active dimers, showing the predicted separation between the N-termini of the two EGF ligands and the average EGF-membrane distance. (**c**) The arrangement of the two intracellular active kinase dimers in the tetramer model. The phosphorylation site Tyr992 (green) of one receptor is positioned close to the active site of a kinase domain (red) in the neighboring dimer.

**Figure 11 mps-02-00012-f011:**
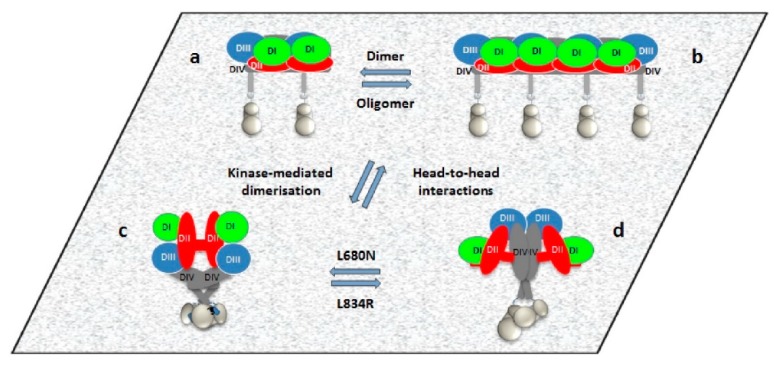
Cartoon models of ligand-free EGFR species on the cell surface (from Zanetti-Domingues et al. (2018) [120]). (**a**,**b**) Autoinhibited ligand-free receptors form dimers and larger oligomers via the extracellular head-to-head interaction. The intracellular domains do not interact; (**c**,**d**) Kinase-mediated dimerization outcompetes the head-to-head interaction and forms two types of receptor dimers existing in equilibrium. The equilibrium is shifted by the kinase-domain mutations L680N and L834R.

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
