# Peer review of "A Brief History of Single-Particle Tracking of the Epidermal Growth Factor Receptor"

_mps, 2019, doi:10.3390/mps2010012_

Round 1

Reviewer 1 Report

The paper submitted by Clark and Martin-Fernandez reviews a series of studies carried out on EGFR by means of SPT techniques, either alone or in combination with FRET and other time-resolved fluorescence microscopy techniques (single-step photobleaching, FLImP).  Although the choice of restricting the review to a specific family of techniques and to a single biological topic might limit its potential audience, I find the subject potentially interesting. 

In my opinion, the manuscript is quite complete but requires some rewriting and reorganization. In fact, I find some parts a bit confusing and some statements about its motivations produce the feeling of an apparent lack of coherence. 

Generally, I think that the paper lacks some quantitative insight, a higher level of detail regarding the techniques (and associated analyses), rigor in the definition of mathematical quantities and concepts associated to types of motion and sometimes a critical comparison of results obtained with different techniques,

Specific comments: 

In the title, the author state that they will review “single molecule imaging of the EGFR”, suggesting that they will present results obtained with several imaging techniques. However, in the abstract and in the introduction, after a brief and generic paragraph on SM imaging, they state that they will exclude from the review “fluorescence microscopy techniques based on single molecule localization” and that they will only focus on SPT. In this view, I guess that the title should be slightly modified to already make clear this choice. Moreover, the authors do not provide a precise motivation for this choice.

It might sound semantic, but SPT is a “fluorescence microscopy technique based on single molecule localization”. I guess that the manuscript will benefit from a clarification of what the authors mean by SPT family, including its possibilities to provide simultaneously to SM tracks, also FRET, polarization, intensity signals.  

The statement “SPT is able to show how fast and how far individual molecules move” is rather generic and not rigorous. In fact, the authors later show that other information (oligomerization, interactions) can be obtained from the same family of techniques. In general, the description of SPT is rather incomplete for a paper meant to partly review a technique. Please consider including a section fully dedicated to a more detailed description of the techniques. 

When describing the results obtained by means of SPT, there is often a vague reference to fundamental concepts without providing a precise definition. For example: saying that MSD curves “plot the distance of a molecule from its starting position against time” is conceptually and dimensionally wrong. 

On the other side, there is no reference to how the single molecule coordinates are obtained from the fluorescence spots (until page 7) and how the trajectories are built. 

“the method has been used to increase our understanding of the receptor’s structure and function” -  This sentence might be misleading, SPT studies cannot report on receptor structure directly. I think the author refer to receptor organization in the cell membrane, i.e. formation of oligomers, or conformational studies involving (also) FRET.

At low-medium expression levels, which do not lead to ligand-independent phosphorylation (typically <105 receptor copies/cell) [15] the EGFR is sufficiently sparse to allow SM imaging of a significant fraction of receptors without overcrowding [16].

I don’t understand the point of the sentence and its relationship with reference [16]. For SPT, if labeling is performed through e.g. antibodies, the concentration of labeled molecules can be controlled and the expression level does not really matter. When labeling with fluorescence proteins receptors are generally overexpressed. 

Figure 1a shows what can be described as “confined”, in which …

I guess there is something missing: confined motion? Confined diffusion?

“Single-step photobleaching” has been often used in experiments that do not involve tracking of the molecule position, therefore cannot be considered a “unique property of SPT”.  

Considering the main objective of the review, it seems that figure 3 is not adding valuable information. In my opinion, it would be better to introduce in the text an explanation of why multicolor SPT is needed, maybe in a technical section (as suggested above).

A single figure can summarize the content of Figure 1, Figure 5 and Figure 10.

In my opinion, the sentence “In this instrument the same area of the sample is imaged simultaneously on the detector at three wavelengths” is not fully explicative for non-experts. The fluorescence emitted in three distinct spectral ranges is optically filtered and collected on different areas of the detector.

The paragraph on PSF fitting comes a bit late since some of experiments described earlier in the ms already use a similar approach. Moreover, I do not fully understand what they mean by “the position of molecules was calculated using a Gaussian mask algorithm, fitting 2D Gaussian curves using iterations of least-square estimators”. Also, if a fit was performed then it is not clear why “Fluorescence intensities were obtained by summing 3 x 3 pixels (100 nm/pixel), and as a control for measurements in cells, the intensities of fluorophores immobilized on glass were also measured”. The latter sentence seems to have no connections with the paragraph where it is inserted.

When discussing the use of QD, there is no mention to blinking.

The sentence “Selection of organic probes according to their hydrophobic properties is also important [16]” at page 8 comes a bit out of the blue. 

Figure 9: please insert a legend or explain in caption.

Author Response

It was our intention for this article to take a rather different approach to the standard techniques review, of which a number are available for single particle tracking (SPT). Instead our goal was to focus on a specific biological system and demonstrate how the development of single particle tracking techniques has advanced research in the field. It is our hope that this will inspire life sciences researchers to consider the application of SPT techniques to their particular research area. Throughout the manuscript we refer to other reviews and specific papers to which readers can turn if they require more detail about specific methods. We believe that, probably due to the title we chose, two of the reviewers were expecting a more conventional techniques review. We have therefore revised the title to more closely reflect the content of the manuscript. In addition, we have revised the introduction to clarify the scope of the article.

We accept that the article would be improved by the insertion of a section describing the techniques used and some of their basic principles, before the review of their biological applications. We have therefore added a new ~ 2 page section “Background to SPT techniques” following the introduction.

Comments and suggestions for Authors

We have addressed the general comments and suggestions above, mainly through the addition of the “Background to SPT Techniques” section.

In the title, the author state that they will review “single molecule imaging of the EGFR”, suggesting that they will present results obtained with several imaging techniques. However, in the abstract and in the introduction, after a brief and generic paragraph on SM imaging, they state that they will exclude from the review “fluorescence microscopy techniques based on single molecule localization” and that they will only focus on SPT. In this view, I guess that the title should be slightly modified to already make clear this choice. Moreover, the authors do not provide a precise motivation for this choice.

We agree with the reviewer and have changed the title to one that we feel is more appropriate.

It might sound semantic, but SPT is a “fluorescence microscopy technique based on single molecule localization”. I guess that the manuscript will benefit from a clarification of what the authors mean by SPT family, including its possibilities to provide simultaneously to SM tracks, also FRET, polarization, intensity signals.

We have revised the introduction to clarify our distinction between techniques that provide images, which we exclude, and the SPT methods that we describe and discuss.

The statement “SPT is able to show how fast and how far individual molecules move” is rather generic and not rigorous. In fact, the authors later show that other information (oligomerization, interactions) can be obtained from the same family of techniques. In general, the description of SPT is rather incomplete for a paper meant to partly review a technique. Please consider including a section fully dedicated to a more detailed description of the techniques. 

When describing the results obtained by means of SPT, there is often a vague reference to fundamental concepts without providing a precise definition. For example: saying that MSD curves “plot the distance of a molecule from its starting position against time” is conceptually and dimensionally wrong. 

On the other side, there is no reference to how the single molecule coordinates are obtained from the fluorescence spots (until page 7) and how the trajectories are built. 

These issues are addressed in the new “Background to SPT techniques” section. We have added some additional detail here. As stated above, we do not intend the article to be a comprehensive and detailed description of the theory of SPT methods (this is available elsewhere), but instead a review that describes the development of SPT methods in the context of a particular biological system.

“the method has been used to increase our understanding of the receptor’s structure and function” -  This sentence might be misleading, SPT studies cannot report on receptor structure directly. I think the author refer to receptor organization in the cell membrane, i.e. formation of oligomers, or conformational studies involving (also) FRET.

We believe the reviewer is using a rather narrow definition of the word “structure” here. The organization and architecture of receptor complexes is “structure”. However, we accept that this might be confusing so we have substituted “structure” with “organization”.

At low-medium expression levels, which do not lead to ligand-independent phosphorylation (typically <105 receptor copies/cell) [15] the EGFR is sufficiently sparse to allow SM imaging of a significant fraction of receptors without overcrowding [16].

I don’t understand the point of the sentence and its relationship with reference [16]. For SPT, if labeling is performed through e.g. antibodies, the concentration of labeled molecules can be controlled and the expression level does not really matter. When labeling with fluorescence proteins receptors are generally overexpressed. 

We agree with the reviewer and have removed this sentence.

Figure 1a shows what can be described as “confined”, in which …

I guess there is something missing: confined motion? Confined diffusion?

We have added the word “motion” here.

“Single-step photobleaching” has been often used in experiments that do not involve tracking of the molecule position, therefore cannot be considered a “unique property of SPT”.  

We have changed “SPT” to “single molecules”.

Considering the main objective of the review, it seems that figure 3 is not adding valuable information. In my opinion, it would be better to introduce in the text an explanation of why multicolor SPT is needed, maybe in a technical section (as suggested above).

We agree that this figure is somewhat out of place with the objective of the review. We have therefore removed it and now refer to multicolor SPT in the “Background to SPT techniques” section. We have included here a reference to a publication in which a diagram of a multicolor TIRF microscope used for SPT experiments can be found.

A single figure can summarize the content of Figure 1, Figure 5 and Figure 10.

We agree that Figure 10 does not add a great deal of additional information so we have removed it. We have also removed Figure 5a. Figure 5b is intended to show the advantages of quantum dots for better distinguishing different types of motion. We introduce this concept at the point where we first discuss their application to SPT of the EGFR. We have therefore retained Figure 5 for this alone.

In my opinion, the sentence “In this instrument the same area of the sample is imaged simultaneously on the detector at three wavelengths” is not fully explicative for non-experts. The fluorescence emitted in three distinct spectral ranges is optically filtered and collected on different areas of the detector.

This is now in the “Background to SPT Techniques” section and we have used the wording suggested by the reviewer.

The paragraph on PSF fitting comes a bit late since some of experiments described earlier in the ms already use a similar approach. Moreover, I do not fully understand what they mean by “the position of molecules was calculated using a Gaussian mask algorithm, fitting 2D Gaussian curves using iterations of least-square estimators”. Also, if a fit was performed then it is not clear why “Fluorescence intensities were obtained by summing 3 x 3 pixels (100 nm/pixel), and as a control for measurements in cells, the intensities of fluorophores immobilized on glass were also measured”. The latter sentence seems to have no connections with the paragraph where it is inserted.

The concept of PSF fitting is now introduced in the “Background to SPT Techniques” section. We agree that the detail above is not particularly informative in the context of the experiment described, so we have simplified the paragraph.

When discussing the use of QD, there is no mention to blinking.

Blinking is now discussed in the “Background to SPT Techniques” section.

The sentence “Selection of organic probes according to their hydrophobic properties is also important [16]” at page 8 comes a bit out of the blue. 

As we now refer to probe selection in “Background to SPT Techniques” we have removed this sentence.

Figure 9: please insert a legend or explain in caption.

Done.

Reviewer 2 Report

The manuscript by Clarke et al. aims at providing a pocket guide to the study of the EGFR by SPT.

While this review gives a good overview, it falls short in providing enough technical background to be helpful as a pocket guide. Instead, it re-narrates quite bit of biological literature but without providing the technological depth to be helpful for a reader that is interested in methods and protocols to study EGFRs by SPT.

Detailed points:

-The difference of oblique and TIRF is not explained. Shading issues with oblique illumination are not discussed. Wavelength-dependent differences in TIRF penetration depth and their potential effect on dual-color SPT are not mentioned.

-A theoretical framework for MSD calculations is missing. Not time, but lag time is used to calculate the MSD. The authors should depict how distances over lag time are calculated and why this leads to much lower accuracy for long lag times. Also while Einstein-Stoked is mentioned later, no discussion can be found on how to calculate the diffusion coefficient nor the anomalous diffusion exponent. Active diffusion is not even mentioned.

-Fig.1d shows a mixture of directed and diffusive behaviour. Track segmentation is not mentioned but would be needed to distinguish modes in the shown track. 

-Fig.3 needs detail to be useful for any reader as to build of adjust a TIRF microscope. How does a "TIRF slider" work? The Optosplit is a commercial solution but no reference is listed. Apertures to project images next to each other are missing in the figure.

-no background on FRET is given, nor are the necessary controls discussed. What kind of FRET analysis was used (radiometric, lifetime...)? What precautions have to be taken...

Fig.5. If the outer ring should represent a confinement, it should be reflected in a saturation in the corresponding MSD track for long time lags. Why do the authors instead show a MSD trace indicating directed motion? Again doing back to track segmentation. What are the solutions/problems?

-Quantum dots: not mention of common problems with QD blinking. How does this complicate SPT?

-Fluorophore stability: No mention of ways to reduce bleaching. What effect does higher peak intensity vs. exposure time have?

-Labelling: There is no discussion on how large probes like antibodies might change single particle behaviour. 

-tracking software: no mention of how developments on SMLM have expanded the capabilities also for SPT. There is much better localisation software now that can be "misused" for SPT.  What about sptPALM?

-no mention on linking software. What are the differences?

-on/off rates: How are binding rates calculated? What are the pitfalls?

-in general: do the authors have permission for all figures used from other publications?

Author Response

It was our intention for this article to take a rather different approach to the standard techniques review, of which a number are available for single particle tracking (SPT). Instead our goal was to focus on a specific biological system and demonstrate how the development of single particle tracking techniques has advanced research in the field. It is our hope that this will inspire life sciences researchers to consider the application of SPT techniques to their particular research area. Throughout the manuscript we refer to other reviews and specific papers to which readers can turn if they require more detail about specific methods. We believe that, probably due to the title we chose, two of the reviewers were expecting a more conventional techniques review. We have therefore revised the title to more closely reflect the content of the manuscript. In addition, we have revised the introduction to clarify the scope of the article.

We accept that the article would be improved by the insertion of a section describing the techniques used and some of their basic principles, before the review of their biological applications. We have therefore added a new ~ 2 page section “Background to SPT techniques” following the introduction.

We believe that some of this reviewer’s requests may result from our title creating confusion regarding the intended scope of the article. We have addressed this point in our general comments above, and believe that the change in title, edits to the introduction, and addition of the “Background to SPT Techniques” section should address this.

The difference of oblique and TIRF is not explained. Shading issues with oblique illumination are not discussed. Wavelength-dependent differences in TIRF penetration depth and their potential effect on dual-color SPT are not mentioned.

We now describe TIRF and inclined illumination in “Background to SPT Techniques”. We believe the additional issues are beyond the scope of this manuscript and are addressed in publications to which we refer.

A theoretical framework for MSD calculations is missing. Not time, but lag time is used to calculate the MSD. The authors should depict how distances over lag time are calculated and why this leads to much lower accuracy for long lag times. Also while Einstein-Stoked is mentioned later, no discussion can be found on how to calculate the diffusion coefficient nor the anomalous diffusion exponent. Active diffusion is not even mentioned.

We have included some additional detail concerning MSD calculation in “Background to SPT Techniques”. The calculation of diffusion coefficient is a complex issue and again we feel this is beyond the scope of the article. We have mentioned the difficulties and refer the reader to a comprehensive study already available.

Fig.1d shows a mixture of directed and diffusive behaviour. Track segmentation is not mentioned but would be needed to distinguish modes in the shown track. 

We now mention this in the article and refer readers to articles in which segmentation is extensively discussed.

Fig.3 needs detail to be useful for any reader as to build of adjust a TIRF microscope. How does a "TIRF slider" work? The Optosplit is a commercial solution but no reference is listed. Apertures to project images next to each other are missing in the figure.

In response to a comment from Reviewer 1 we have removed Fig. 3 so this comment no longer applies.

no background on FRET is given, nor are the necessary controls discussed. What kind of FRET analysis was used (radiometric, lifetime...)? What precautions have to be taken...

We now introduce FRET in “Background to SPT Techniques”. Again, we feel that the reviewer is asking for detail that is beyond the intended scope of the article. We refer to articles in which much more detail on spFRET can be found if the reader requires it.

Fig.5. If the outer ring should represent a confinement, it should be reflected in a saturation in the corresponding MSD track for long time lags. Why do the authors instead show a MSD trace indicating directed motion? Again doing back to track segmentation. What are the solutions/problems?

We have removed Fig. 5a in response to a comment from Reviewer 1. The MSD trace remaining is intended to show the advantages of using QDs to obtain longer tracks, enabling directed motion to be distinguished better from random diffusion. We believe it is appropriate to show this plot at this point in the manuscript.

Quantum dots: not mention of common problems with QD blinking. How does this complicate SPT?

This is now addressed in “Background to SPT Techniques”.

Fluorophore stability: No mention of ways to reduce bleaching. What effect does higher peak intensity vs. exposure time have?

We refer to probe stability in “Background to SPT Techniques”, and refer to a comprehensive review of fluorophore photophysics (Ha & Tinnefeld, 2012) in which the issue is discussed in detail.

Labelling: There is no discussion on how large probes like antibodies might change single particle behaviour

tracking software: no mention of how developments on SMLM have expanded the capabilities also for SPT. There is much better localisation software now that can be "misused" for SPT.  What about sptPALM?

no mention on linking software. What are the differences?

on/off rates: How are binding rates calculated? What are the pitfalls?

Again, we now refer to all these issues in the new section of the manuscript. References to other publications are provided so readers can obtain additional information/detail if they wish.

in general: do the authors have permission for all figures used from other publications?

Permission has been obtained and evidence for this provided.

Reviewer 3 Report

This manuscript by Clark and Martin-Fernandez is a fine review of the single-molecule fluorescence techniques used to study EGFR. It is a well-written and thorough review.

The phrase “pocket guide” within the manuscript title suggests that the manuscript will include guidance about how to do future experiments. Instead, this manuscript provides a review of what has been done previously.

The introduction would benefit from including an outline of the various fluorescence techniques to be discussed and a brief comment of the unique information each technique provides.

Figure 2a and 2b could use an x-axis label. Some of the meaning of Fig 2 is lost on the readers because it is introduced before the discussion of how these techniques work. For example, it is not clear how the I vs. t data is integrated to find the single diffuser brightnesses. The paper may broadly benefit from talking about the methods before discussing results.

Figure 6d misrepresents the localization precision and the relative magnitude of r. Figure 6c, d, and f would benefit from having units indicated on the x-axes.

The legend of Figures 9 and 10 would benefit for better clarification. For example, in Figure 9, Is the structure shown a tetramer where each protein is shown in a unique color? Is the point of Figure 10 to show the variation in MSD vs. Deltat plots observed or are the panels of Figure 10 well correlated with oligomerization state? As written, Figure 10 seems to be very similar to Figure 1, and perhaps these two figures could be combined.

On page 21, the sentence “We have already discussed how SPT can be used to investigate the dynamics of molecular interactions in cells, including measuring the strength of interaction by determining on and off rates.” may be misleading. There was a minimal discussion of on or off-rates previously in the paper.

The manuscript would benefit from a more significant discussion of the future directions of both the fluorescence methods and the critical properties of EGFR that are yet to be measured.

The commas in this sentence aren’t correct. “Proteins, involved in cell signaling are believed to partition into lipid rafts, suggesting a possible role for rafts in the regulation of signal transduction [39,40].”

There should usually be a hyphen in “single molecule localization,” “single particle tracking,” or “single point emitters.”

Author Response

It was our intention for this article to take a rather different approach to the standard techniques review, of which a number are available for single particle tracking (SPT). Instead our goal was to focus on a specific biological system and demonstrate how the development of single particle tracking techniques has advanced research in the field. It is our hope that this will inspire life sciences researchers to consider the application of SPT techniques to their particular research area. Throughout the manuscript we refer to other reviews and specific papers to which readers can turn if they require more detail about specific methods. We believe that, probably due to the title we chose, two of the reviewers were expecting a more conventional techniques review. We have therefore revised the title to more closely reflect the content of the manuscript. In addition, we have revised the introduction to clarify the scope of the article.

We accept that the article would be improved by the insertion of a section describing the techniques used and some of their basic principles, before the review of their biological applications. We have therefore added a new ~ 2 page section “Background to SPT techniques” following the introduction.

The phrase “pocket guide” within the manuscript title suggests that the manuscript will include guidance about how to do future experiments. Instead, this manuscript provides a review of what has been done previously.

The introduction would benefit from including an outline of the various fluorescence techniques to be discussed and a brief comment of the unique information each technique provides.

These issues have been addressed by a change of title and the addition of a new section (see general response above).

Figure 2a and 2b could use an x-axis label. Some of the meaning of Fig 2 is lost on the readers because it is introduced before the discussion of how these techniques work. For example, it is not clear how the I vs. t data is integrated to find the single diffuser brightnesses. The paper may broadly benefit from talking about the methods before discussing results.

X-axis labels have been added. The other point is addressed by the inclusion of the new “Background to SPT Techniques” section.

Figure 6d misrepresents the localization precision and the relative magnitude of r. Figure 6c, d, and f would benefit from having units indicated on the x-axes.

It is unclear to us what the reviewer means in the first sentence. The figure is reproduced from a previous publication in Nature Communications. Regarding the second point, the relevant parts of the figure illustrate the general principles behind the FLImP method, so there are no specific units.

The legend of Figures 9 and 10 would benefit for better clarification. For example, in Figure 9, Is the structure shown a tetramer where each protein is shown in a unique color? Is the point of Figure 10 to show the variation in MSD vs. Deltat plots observed or are the panels of Figure 10 well correlated with oligomerization state? As written, Figure 10 seems to be very similar to Figure 1, and perhaps these two figures could be combined.

We have added to the legend of Figure 9 for clarification. We decided that Figure 10 does not add any significant information to that already given in Figure 1, so we have removed it.

On page 21, the sentence “We have already discussed how SPT can be used to investigate the dynamics of molecular interactions in cells, including measuring the strength of interaction by determining on and off rates.” may be misleading. There was a minimal discussion of on or off-rates previously in the paper.

We have revised this sentence and referred to where we mention on and off rates in the new section.

The manuscript would benefit from a more significant discussion of the future directions of both the fluorescence methods and the critical properties of EGFR that are yet to be measured.

We have expanded the final section of the paper to discuss these issues further.

The commas in this sentence aren’t correct. “Proteins, involved in cell signaling are believed to partition into lipid rafts, suggesting a possible role for rafts in the regulation of signal transduction [39,40].”

Corrected

There should usually be a hyphen in “single molecule localization,” “single particle tracking,” or “single point emitters.”

Hyphens added where appropriate.

Round 2

Reviewer 1 Report

The authors have satisfactorily addressed most of my comments.

Minor comments:

Page 3: missing word in  “Detection of the fluorescence signals is usually ??? by either an electron-multiplying CCD (EMCCD) or more recently a Scientific CMOS (sCMOS) camera.”

Page 4: extra/missing word in “Typically in SPT experiments the localization precision an range from a few nanometers to a few tens of nanometers.”

In the methodological section (page 4), the authors should cite "Nature Methods volume 11, pages 281–289 (2014)"

On page 31, when discussing new approaches for data analysis, the authors could mention e.g. "PNAS, 115 9026-9031 (2018)".

Author Response

Page 3: missing word in  “Detection of the fluorescence signals is usually ??? by either an electron-multiplying CCD (EMCCD) or more recently a Scientific CMOS (sCMOS) camera.”

We believe the original sentence is grammatically correct, but to avoid confusion we have altered it slightly.

Page 4: extra/missing word in “Typically in SPT experiments the localization precision an rangefrom a few nanometers to a few tens of nanometers.”

This is a typo and should read "can range...". We have corrected this.

In the methodological section (page 4), the authors should cite "Nature Methods volume 11, pages 281–289 (2014)"

Now cited.

On page 31, when discussing new approaches for data analysis, the authors could mention e.g. "PNAS, 115 9026-9031 (2018)".

Now cited.

Reviewer 2 Report

I thank the authors for including more background in this review and also changing the title to make the intention of this manuscript clearer.

However, I just realised that Figure 1 looked too familiar. Indeed, this figure seems to be taken and slightly modified from Figure 4 in Bacher et al. 2004 (https://bmcmolcellbiol.biomedcentral.com/articles/10.1186/1471-2121-5-45).

Since the authors do not state that they copied it from this source but instead slightly modified it, I have to suspect plagiarism.

I therefore forward this to the editor for a decision on how to proceed.

Author Response

However, I just realised that Figure 1 looked too familiar. Indeed, this figure seems to be taken and slightly modified from Figure 4 in Bacher et al. 2004 (https://bmcmolcellbiol.biomedcentral.com/articles/10.1186/1471-2121-5-45).

Thanks to the reviewer for pointing this out. In the manuscript we reproduce or adapt a number of figures and have acknowledged the sources appropriately (and obtained permission where needed). Through an oversight we omitted to do this for Figure 1. We have now inserted a reference to the source in the figure caption. As the article from which the figure was adapted is open source, permission from the publisher is not required.